# The regulation landscape of MAPK signaling cascade for thwarting *Bacillus thuringiensis* infection in an insect host

Zhaojiang Guo[1,Ⓤ]*, Shi Kang[1,Ⓤ], Qingjun Wu[1], Shaoli Wang[1], Neil Crickmore[2], Xuguo Zhou[3], Alejandra Bravo[4], Mario Soberón[4], Youjun Zhang[1]*

**1** Department of Plant Protection, Institute of Vegetables and Flowers, Chinese Academy of Agricultural Sciences, Beijing, China, **2** School of Life Sciences, University of Sussex, Brighton, United Kingdom, **3** Department of Entomology, University of Kentucky, Lexington, Kentucky, United States of America, **4** Departamento de Microbiología Molecular, Instituto de Biotecnología, Universidad Nacional Autónoma de México, Cuernavaca, Morelos, Mexico

Ⓤ These authors contributed equally to this work.

* guozhaojiang@caas.cn (ZG); zhangyoujun@caas.cn (YZ)

**Data Availability Statement:** All relevant data are within the manuscript and its Supporting Information files. The final cloned full-length cDNA sequences of P. xylostella MAPK cascade genes

## Abstract

Host-pathogen interactions are central components of ecological networks where the MAPK signaling pathways act as central hubs of these complex interactions. We have previously shown that an insect hormone modulated MAPK signaling cascade participates as a general switch to *trans*-regulate differential expression of diverse midgut genes in the diamondback moth, *Plutella xylostella* (L.) to cope with the insecticidal action of Cry1Ac toxin, produced by the entomopathogenic bacterium *Bacillus thuringiensis* (Bt). The relationship between topology and functions of this four-tiered phosphorylation signaling cascade, however, is an uncharted territory. Here, we carried out a genome-wide characterization of all the MAPK orthologs in *P. xylostella* to define their phylogenetic relationships and to confirm their evolutionary conserved modules. Results from quantitative phosphoproteomic analyses, combined with functional validations studies using specific inhibitors and dsRNAs lead us to establish a MAPK "road map", where p38 and ERK MAPK signaling pathways, in large part, mount a resistance response against Bt toxins through regulating the differential expression of multiple Cry toxin receptors and their non-receptor paralogs in *P. xylostella* midgut. These data not only advance our understanding of host-pathogen interactions in agricultural pests, but also inform the future development of biopesticides that could suppress Cry resistance phenotypes.

## Author summary

The MAPK signaling pathways are pivotal for triggering host immunity against pathogens during their intricate environmental interactions. Moreover, extensive studies have demonstrated that dysfunctions in these pathways are associated with serious diseases in plants and mammals. Despite its importance, the four-tiered signaling cascades of these signaling pathways to exert functions still remain as a "black box" and have seldom been defined in

were deposited in the GenBank database (Accession nos. MN211342-MN211357).

**Funding:** This work was supported by the National Natural Science Foundation of China (31701813 and 32022074 to ZG; 31901917 to SK; 31630059 to YZ; funder website: http://www.nsfc.gov.cn/english/site_1/index.html), the Beijing Key Laboratory for Pest Control and Sustainable Cultivation of Vegetables and the Science and Technology Innovation Program of the Chinese Academy of Agricultural Sciences (CAAS-ASTIP-IVFCAAS) to YZ. The funders had no role in study design, data collection and analysis, decision to publish, or preparation of the manuscript.

**Competing interests:** The authors have declared that no competing interests exist.

insects, especially in non-model agricultural insect pests. Recently, we have discovered that an insect hormone activated MAPK signaling cascade is pivotal to overcome the toxic action of *Bacillus thuringiensis* (Bt) toxin thereby resulting in high-level Bt Cry1Ac resistance in its insect host, the "super pest" diamondback moth, *Plutella xylostella* (L.). Here, we further deciphered the three underlying activation routes for the complex four-tiered MAPK signaling modules (including MAP4K4-Raf-MAP2K1-ERK, MAP4K4-MAP3K7-MAP2K4-JNK and MAP4K4-MAP3K7-MAP2K6-p38) to orchestrate the differential expression of multiple midgut genes and confer high-level resistance to the Bt Cry1Ac toxin in *P. xylostella*. Our study provides the first comprehensive mechanistic insights into the four-tiered MAPK signaling cascades involved in insect resistance to Bt toxins, and also provides a platform for uncovering the topology and functions of these complex intracellular immune signaling pathways in non-model agricultural insects.

## Introduction

Plants and animals live in an environment teeming with multiple pathogens, and have thus evolved strategies to withstand pathogen attack. Defense responses through activation of signaling pathways, from pathogen recognition to the induction of immune responses, are crucial in host-pathogen interactions [1,2]. Typically, the mitogen-activated protein kinase (MAPK) signaling pathways play crucial roles in the arms race between host and pathogen [3]. The MAPK signaling pathways are evolutionarily conserved modules in all eukaryotes and are characterized by multi-tiered phosphorylation cascades composed of MAPKKK kinase (MAP4K), MAPKK kinase (MAP3K), MAPK kinase (MAP2K) and MAPK [4,5]. The contribution of MAPK signaling cascades in controlling cellular responses to a wide assortment of stimuli and in regulating cellular processes from gene expression to cell death is well established [6].

Host-pathogen interactions are not only restricted to plants and mammals. Insects, the largest group of animals on earth, can be infected by a wide range of pathogens, including bacteria, fungi, viruses and parasites [7]. *Bacillus thuringiensis* (Bt), a gram-positive entomopathogenic bacterium, produces diverse pore-forming toxins (PFTs) such as Cry toxins as virulence factors to specifically kill their insect hosts [8], in a fashion similar to some human pathogenic bacteria [9]. Biopesticide formulations and transgenic crops based on Bt insecticidal toxins are widely adopted in pest control worldwide, providing tremendous ecological and socio-economic benefits [10–17]. However, insect pests have developed intricate strategies to counteract the detrimental effects caused by Bt toxins, thus, defining these molecular mechanisms evolved in insect hosts to counteract Bt infection is pivotal to establish successful strategies to counter insect resistance to Bt toxins [18–23].

The mode of action of Bt Cry toxins involves a complex multi-step process in which toxin-receptor interactions are crucial [24]. Alterations in midgut receptors disrupting these binding interactions with Cry toxins generally correlate with high levels of resistance in diverse insect pests [22,25]. The diamondback moth, *Plutella xylostella* (L.), is a cosmopolitan pest that was the first insect recognized to develop field-evolved resistance to Bt biopesticides [26,27]. The release of its whole genome sequence [28] has made it an excellent model to define the molecular basis of host-pathogen interactions. Previously, field-evolved resistance to the Bt Cry1Ac toxin in *P. xylostella* was associated with both a *cis*-mutation in an ABC transporter gene [29] and a *trans*-regulatory mechanism involving a hormone activated MAPK signaling pathway that altered the expression of midgut genes encoding glycosyl-phosphatidylinositol (GPI)-

anchored alkaline phosphatase (ALP), and aminopeptidase N (APN) proteins as well as transmembrane ABC transporter proteins [30–34]. The fact that resistance in *P. xylostella* has been linked with multiple and different mechanisms has motivated us to further gain a clear understanding of the underlying interactions. In this work, we decided to further analyze the role of MAPK signaling pathways in the developed resistance to Bt toxins [35].

The MAPK signaling pathways have been extensively studied in plants and mammals [5,36]. In particular, it was previously shown that the MAPK signaling pathways can be responsive to diverse PFTs in mammals, *Drosophila* and *Caenorhabditis elegans* [9,37,38]. Moreover, it was also shown that these pathways also play important roles in the response against Bt Cry toxins in insects [30,32,39–41]. However, the full repertoire and function of this immune defensive response in insects, particularly in non-model insects of agricultural relevance such as *P. xylostella*, is currently not deeply understood. In this study, we conducted a genome-wide identification and characterization of the four-tiered MAPK signaling cascades in *P. xylostella*. We uncovered their topological structure and functional mechanism of the cascades involved in directing the expression of downstream midgut genes repertoire leading to effective Cry1Ac resistance in the absence of fitness costs. The model that resulted from this study indicated a clear possibility of developing products that could suppress the resistance phenotype.

## Results

### Genome-wide characterization of MAPK kinases in *P. xylostella* and other arthropods

Our previous studies have confirmed a crucial role of the MAPK signaling cascades in Cry1Ac resistance in *P. xylostella* [30,32], in this study, the potential members of these cascades were further identified in *P. xylostella* (S1 Table) and in 10 other arthropod species (S2 Table). Using currently available transcriptome and genome databases, a total of 17 MAPK orthologs in *P. xylostella* were found *in silico* and their genes were cloned (S1 Table, S1A Fig). These genes displayed polymorphism in both sequence length and exon number and were distributed between different scaffolds of the *P. xylostella* genome (S1B and S2 Figs).

Similar MAPK orthologs were found among other arthropods (Fig 1A) and the analysis of the non-synonymous/synonymous mutation ratio (Ka/Ks values) of each gene allowed us to estimate values between 0 and 0.25 implying that these genes were under purifying or stabilizing selection (Fig 1B), and suggesting that these signaling pathways are evolutionary conserved modules that have diverged very little among the analyzed arthropods. In the hierarchical fashion of MAPK cascades, MAP3K exhibits higher diversity, not only in distribution (Fig 1A) but also in their protein sequences (Figs 1C and S3), when compared with MAP4K, MAP2K and MAPK (Figs 1C and S3). We also found that the *mos* gene is missing in the 15 lepidopteran insects analyzed including *P. xylostella* (S4 Fig). The TAO kinase can phosphorylate MAP2Ks, a hallmark of a MAP3K, but the primary sequence of TAO is more closely related to MAP4K in the phylogenetic tree (Fig 1C). The lack of a close relationship between TAO and other MAP3Ks was previously observed [42]. In the MAPK group, MAPK15 was absent in four of the 15 species analyzed (Fig 1A), these data and the low protein sequence similarity among MAPK15 and other MAPKs (Figs 1C and S3) suggest that MAPK15 may not be a classical MAPK.

### Expression atlas of the MAPK cascade genes in *P. xylostella*

Currently available RNA-seq data downloaded from Sequence Read Archives (SRA) of GenBank allowed us to search for these sequences and found out that these proteins of MAPK

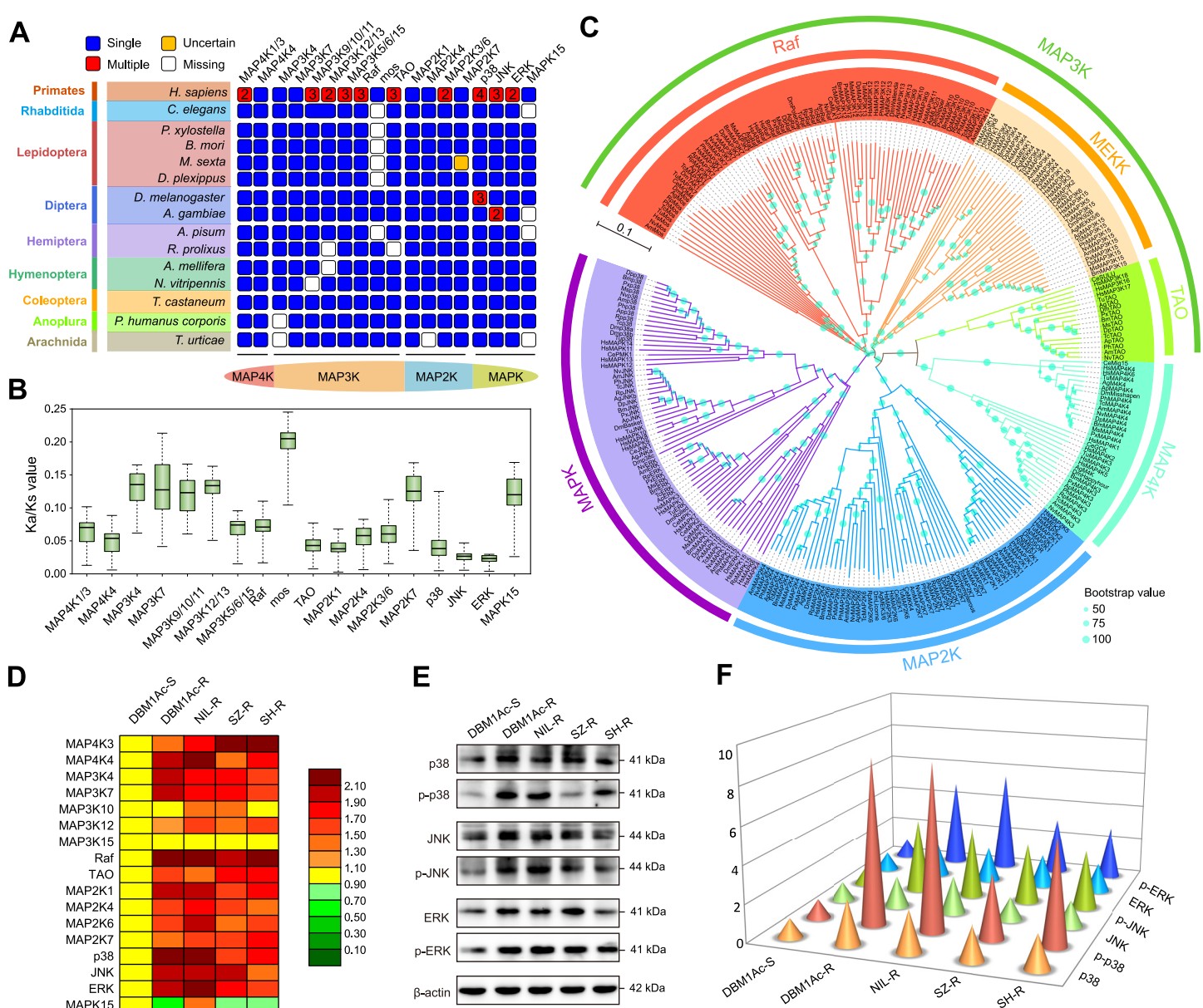

**Fig 1. Genome-wide identification and characterization of MAPK cascade genes in *P. xylostella*.** (A) Distribution of MAPK cascade genes among *H. sapiens*, *C. elegans* and 13 arthropod species. (B) The Ka/Ks values of MAPKs from 13 arthropod species. (C) Phylogenetic tree of MAPKs from *H. sapiens*, *C. elegans* and 13 arthropod species. MAP3Ks include three clades: Raf, MEKK and TAO. (D) Heatmap showing the relative expression levels of MAPK cascade genes in midgut tissues of fourth-instar *P. xylostella* larvae from the different strains as determined by qPCR analysis. The color of each rectangle denotes the relative expression level of each gene expressed as mean of fold changes relative to the control DBM1Ac-S larvae. Red and green colors indicate up- and down-regulation respectively, while yellow indicates no significant expression variation. (E and F) Western blot analyzes of both total protein and protein phosphorylation levels of p38, JNK and ERK in midgut tissues of fourth-instar larvae from the different strains. The quantification (F) of representative blots (E) using ImageJ 1.51 from three biological replicates is shown. All the control larval samples used in qPCR and western blot analyses (Panels D-F) were without Cry1Ac exposure to quantify the basal expression of MAPK cascade genes.

cascade pathways are expressed in all four developmental stages of *P. xylostella*, and are fairly evenly distributed in six different adult tissues. Also that *PxMAP4K4* and *Pxp38* exhibited higher expression levels than the others during fungal infection (S5A–S5D Fig).

Then, we further analyzed the relative expression levels of *Pxp38*, *PxJNK* and *PxERK* genes by qPCR in the susceptible DBM1Ac-S strain. As expected, their expression was detected in all

the developmental stages, and in all tested tissues of fourth-instar larvae from DBM1Ac-S strain (S6 Fig). More variation was observed between developmental stages than between tissues. When we compared the expression of the 17 MAPK genes mentioned above among four Cry1Ac resistant strains and the susceptible one, the transcript levels of most of the selected MAPK cascade genes (especially *PxMAP4K4*, *PxRaf*, *Pxp38*, *PxERK*) were up-regulated in the midgut tissues of all resistant strains compared to the susceptible strain DBM1Ac-S (Fig 1D). With evidence that genes encoding these components of the signaling cascades were up-regulated in the resistant strains, we decided to use western blot assays to analyze the total protein expression and the phosphorylation levels of the three key downstream MAPK of the different pathways (p38, JNK and ERK) (Fig 1E). In the Cry1Ac resistant strains, the relative protein abundance of p38, JNK and ERK was higher than in the susceptible strain, moreover, their phosphorylated protein levels were also markedly increased ($P < 0.05$; Duncan's test; n = 3) (Fig 1E and 1F).

## The MAPK cascades associated with Cry1Ac resistance in *P. xylostella*

MAP4K4 was previously shown to be involved in the mechanism of resistance to Bt Cry1Ac toxin in the *P. xylostella* strain NIL-R [30,32] and the above data suggested a role for the p38, JNK and ERK MAPKs. Here, we aimed to further identify components of the phosphorylation cascade that may be associated with this resistance phenotype. Thus, global quantitative phosphoproteomic analyses in fourth-instar larvae from both susceptible DBM1Ac-S and its near-isogenic resistant NIL-R strains were performed (Fig 2A). In total, 1652 phosphorylated peptides derived from 846 proteins were identified. We found that 716 phospho-peptides from 547 proteins were quantifiable (S5 Table), revealing that 115 phospho-peptides increased and 75 phospho-peptides decreased in NIL-R compared to DBM1Ac-S (Fig 2B). GO and KEGG analyses indicated that these differentially phosphorylated proteins were involved in a number of different metabolic processes and signaling pathways (Fig 2C and 2D).

Regarding the MAPK components identified as differentially phosphorylated in the NIL-R strain, we found that MAP3K7, TAO, MAP2K6, ERK and p38 showed increased phosphorylation in the resistant strain (Fig 2B), but that the phosphorylation level of MAP4K3, MAP3K4, MAP3K12, MAP3K15 and MAP2K7 did not change between the two strains (S5 Table). To identify the potential downstream effectors of these kinases we built a connectivity network of differentially phosphorylated proteins to illustrate the possible interactions among MAPK members and other phospho-proteins (Fig 2E).

## Effect of MAPK silencing on host midgut gene expression and Cry1Ac susceptibility

Given the likelihood that a common resistance mechanism exists in the four resistant strains [30,32], we determined whether or not there was a causal link between increased expression/phosphorylation of these key kinases and the observed resistance phenotypes. Thus, RNAi was performed on the NIL-R strain to silence the expression of those MAPK cascade genes shown to be differentially phosphorylated in the phosphoproteome screen (Fig 2B) as well as others identified as being overexpressed in all the resistant strains (Fig 1D). qPCR demonstrated that RNAi selectively reduced the expression of each target gene (Fig 3A). The effect of RNAi of these 11 target kinase genes on the phosphorylation of the downstream effector proteins p38, ERK and JNK was determined by western blot analysis of each silenced larva. These data indicated that MAP2K6 was involved in the activation of p38; MAP2K1 in the activation of ERK; and that both MAP2K4 and MAP2K7 could activate JNK (Figs 3B and 3C and S7). Upstream of these MAPKs, Raf was involved in ERK activation, TAO in JNK activation and MAP3K7 in

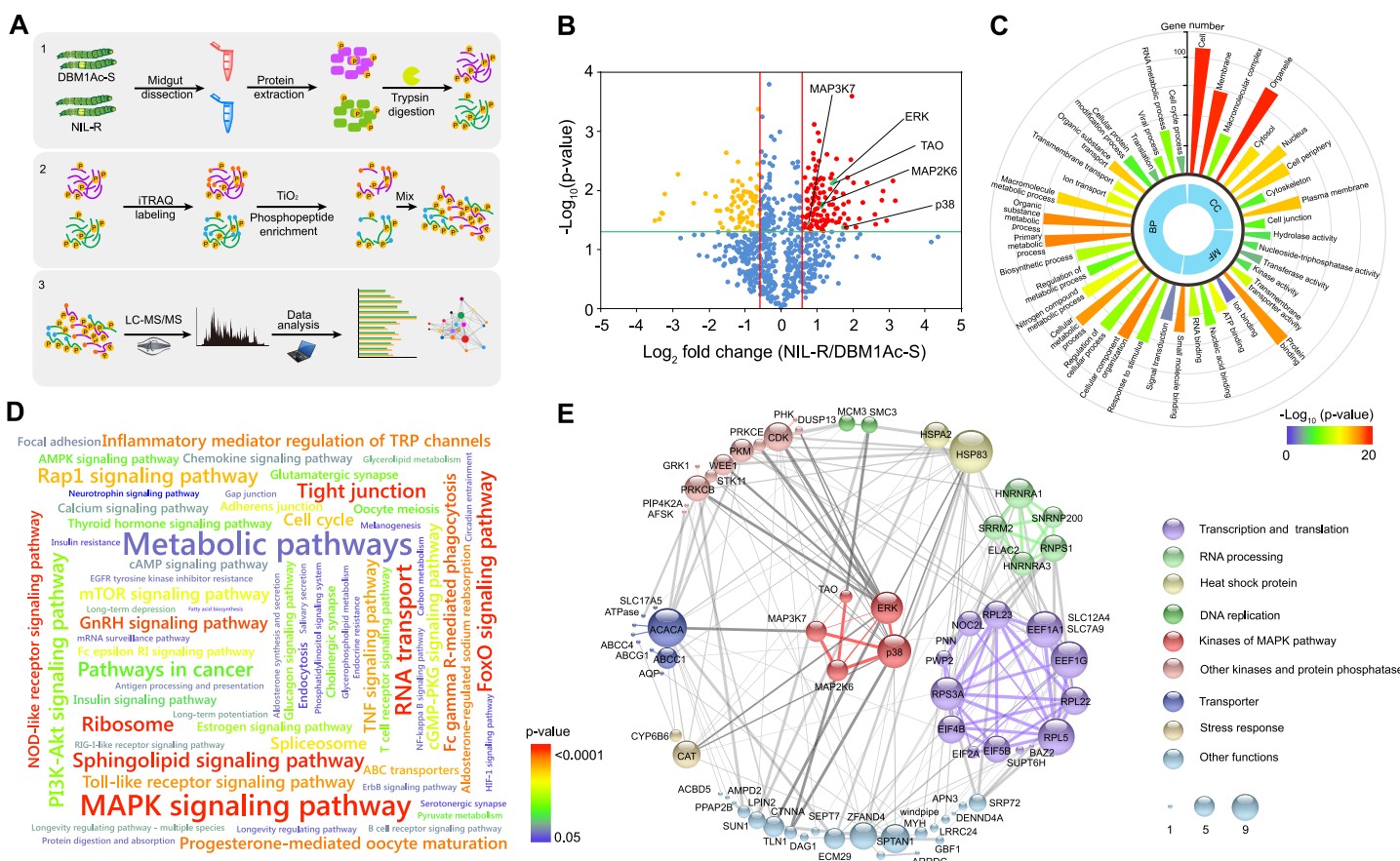

**Fig 2. Quantitative phosphoproteomic profiling for differentially phosphorylated proteins and their interaction relationships in Cry1Ac susceptible and resistant *P. xylostella* strains.** (A) Workflow of the experimental procedure for iTRAQ global phosphoproteome analysis. All the control larval samples used in phosphoproteome analysis were without Cry1Ac exposure to detect basal differentially phosphorylated proteins between resistant and susceptible strains. (B) Volcano plot of the abundance changes of qualified phosphoproteins in DBM1Ac-S versus NIL-R strains. Each dot represents a phospho-peptide. Average phospho-peptide expression ratio of three biological replicates (log 2 transformed) was plotted against p-value by t-test (−log 10 transformed). Cutoff of P = 0.05 and 1.5-fold change were denoted by green and red lines, respectively. Some of the significantly up-regulated MAPK cascade kinases (P ≤ 0.05, fold change ≥ 1.5) are highlighted. (C) GO term enrichment for the significantly changed phosphoproteins. (D) KEGG pathway enrichment for the significantly changed phosphoproteins. Enriched KEGG pathways are visualized as a word cloud. The size of the word corresponds to the gene abundance in that category. (E) Connectivity network of significantly changed phosphoproteins. The original protein-protein interaction data comes from the STRING database. The size of the circles denotes the frequency of interactions. The thickness of the lines denotes the score of the interactions. Interactions among MAPK cascade kinases are shown by red lines, while interactions among MAPK cascade kinases and other proteins are shown by dark grey lines, all other interactions are shown in light grey lines.

both p38 and JNK activation. Finally, the data confirmed that MAP4K4 was involved in the activation of all three key MAPKs.

Individual and combinational RNAi was conducted to silence the expression of key MAPK cascade genes and qPCR were undertaken to establish the effect of silencing these kinases on the expression of those midgut genes related to Cry1Ac resistance in the NIL-R strain (Fig 3D). The results showed that the individual silencing of most of the MAPKs analyzed induced changes in expression of the selected midgut proteins (ALP, APNs and ABC transporters) (Fig 3D). The exceptions were TAO and MAP2K7, since RNAi silencing of these MAPK cascade genes had little or no effect on the expression of the analyzed ALP, APNs or ABC transporters (Fig 3D). Using these expression data, we performed hierarchical clustering and principal component analyses (PCA) (Fig 3E) to establish the potential signaling pathways. This analysis was consistent with that described above (Fig 3B and 3C), indicating that MAP4K4 silencing gave similar results as silencing the three key MAPKs (p38, JNK and ERK). Links were also

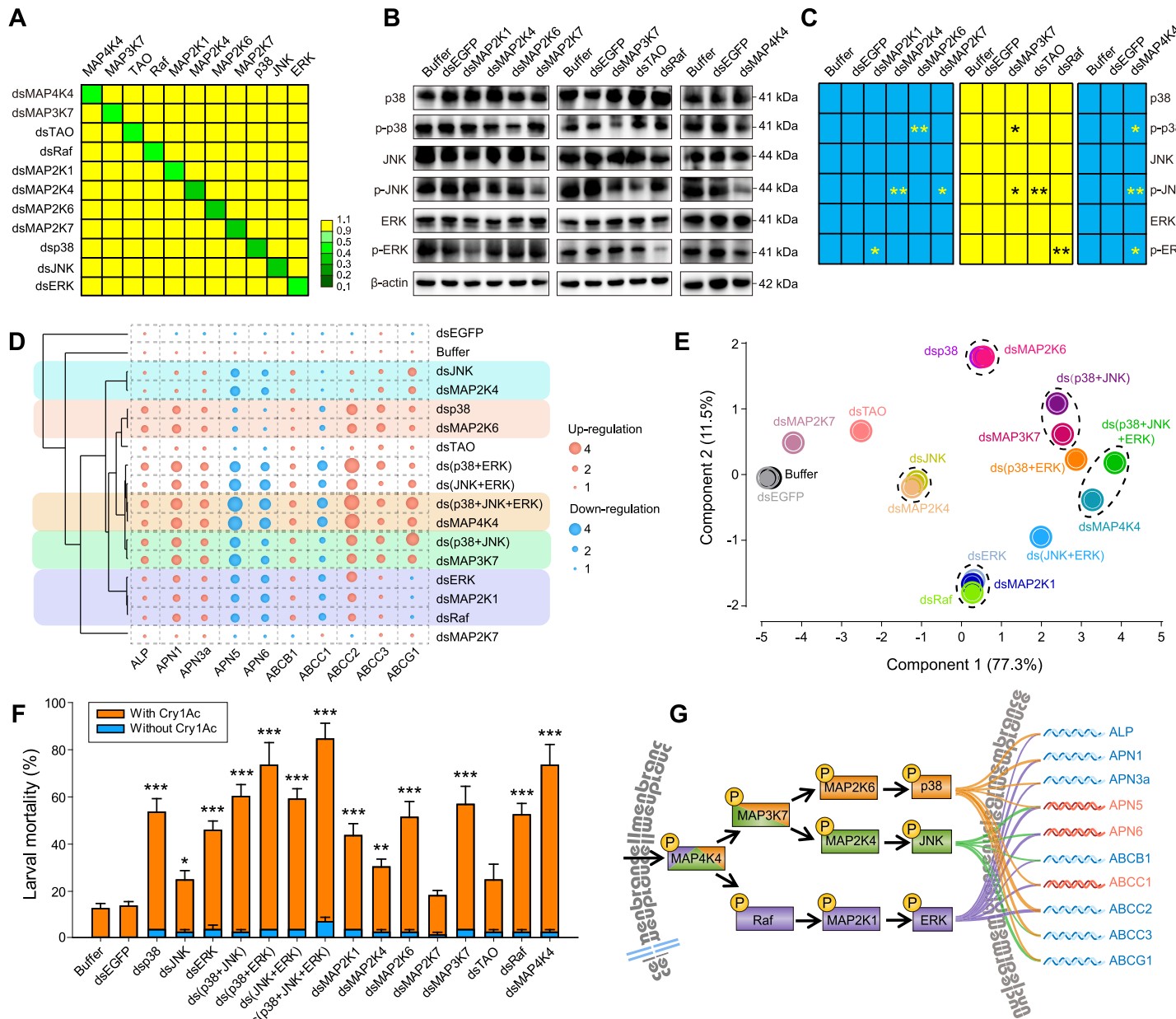

**Fig 3. Identification of the MAPK cascades involved in Cry1Ac resistance.** (A) The relative expression of different MAPK genes at 48 h post-RNAi. The color-coded expression data is calculated relative to the transcript level for each gene at time 0 h. (B) Western blot analysis of total protein and protein phosphorylation levels of p38, JNK and ERK in resistant NIL-R larvae microinjected with dsRNA targeting MAP2Ks, MAP3Ks or MAP4K. Representative blots are shown. The β-actin protein was analyzed as an internal loading control. (C) Quantification of western blots using ImageJ 1.51 from three biological replicates. *P < 0.05 and **P < 0.01 by Duncan's test compared to the controls. (D) Hierarchical clustering of RNAi treated strains analyzing the expression level of resistance-related midgut genes (ALP, APNs and ABC transporters genes). The expression data are from three biological replicates and four technical repeats. (E) Principal component analysis (PCA) of the indicated RNAi-treated strains analyzing the expression level of the target resistance-related midgut genes. (F) Susceptibility to 1000 mg/L Bt Cry1Ac protoxin of NIL-R larvae after RNAi. Data in figures are means and standard errors from three biological replicates. *P < 0.05, **P < 0.01 and ***P < 0.001 by Duncan's test compared to the controls. (G) Schematic representation of the gene regulation landscape of midgut proteins, Cry1Ac-receptors and non-receptor paralogs, involved in Bt Cry1Ac resistance by the activated MAPK signaling cascade pathways. The target downstream midgut resistance-related genes with red color indicated that they were up-regulated, while those with blue color indicated that they were down-regulated.

suggested between MAP3K7 and p38+JNK; between Raf, MAP2K1 and ERK; between MAP2K4 and JNK and between MAP2K6 and p38.

Based on our understanding of the role of the selected midgut proteins (ALP, APNs and ABC transporters) in the mechanism of action of Cry1Ac against *P. xylostella* [30–33], we would expect that the observed expression differences in Fig 3D would correlate with changes in the susceptibility of the silenced larvae to this toxin. To test this hypothesis, bioassays were done on strains in which the different kinases were silenced by RNAi (Fig 3F). The data confirmed that silencing the expression of TAO or MAP2K7 did not alter Cry1Ac susceptibility, which is consistent with the fact that silencing these proteins had little effect on the expression of the midgut proteins of interest. In contrast, reducing the expression of other nine kinases resulted in increased susceptibility of the resistant strain to a greater or lesser extent, supporting the concept that toxin susceptibility was directly related to the expression levels of these midgut Cry1Ac receptors. Our data indicate that among the three key MAPK pathways, the JNK pathway was less important to modulate expression of midgut Cry1Ac receptors than the p38 or ERK pathways (Fig 3F). Our previous work suggested that reduced expression of some midgut functional receptors in resistant insects was associated with increased expression of functional non-receptor paralogs to minimize fitness costs [30,32]. The results presented here agree with those previous data since functional Cry1Ac receptors (PxmALP, PxAPN1, PxAPN3a, PxABCB1, PxABCC2, PxABCC3 and PxABCG1) were all up-regulated in the RNAi-silenced strains, whereas the non-receptor paralogs (PxAPN5, PxAPN6 and PxABCC1) were all down-regulated. With these data, we were able to delineate a diagram for MAPK signaling pathways in the Bt Cry1Ac resistance mechanism in *P. xylostella* (Fig 3G).

## Modulating susceptibility to Cry1Ac toxin with MAPK inhibitors

According to the proposed MAPK signaling pathways described above, we hypothesized that blocking activation of the signaling cascade with specific inhibitors should alter the susceptibility of the resistant NIL-R strain. This was tested by using commercially available inhibitors of p38, JNK and ERK. The use of phosphorylation-specific antibodies confirmed the specific activity of the selected inhibitors, both when applied individually or in combination (Figs 4A, 4B, S8 and S9). Bioassays confirmed that blocking these pathways in the resistant NIL-R strain with inhibitors resulted in a significant increase in the susceptibility to Cry1Ac toxin (Fig 4C). As with the RNAi experiments, inhibition of JNK action had less effect than interfering with p38 or ERK action. Gene expression analysis was finally used to establish the effect of the inhibitors on expression of the selected midgut genes (Fig 4D and 4E). The results were consistent with those obtained in the RNAi experiments supporting the roadmap shown in Fig 3G.

## Discussion

Various organisms including plants, insects and mammals, are continuously engaged in a co-evolutionary struggle against their pathogens. The outcomes of host-pathogen interactions are essential for human activities, as they can have significant impacts on healthcare and agricultural systems [2,43]. In the ongoing battle between hosts and pathogens, the hosts have adapted their capacity to establish immunity strategies to defend against pathogen infections, and pathogens have developed strategies to overcome these defense responses [1,2]. MAPK signaling cascades have proved to be hubs for triggering diverse immune responses in these host-pathogen interactions [3].

In mammals, it is well established that after recognizing conserved microbial elicitors called pathogen-associated molecular patterns (PAMPs), pattern recognition receptors (PRRs) can activate MAPK signaling pathways, which are part of both innate and adaptive immune

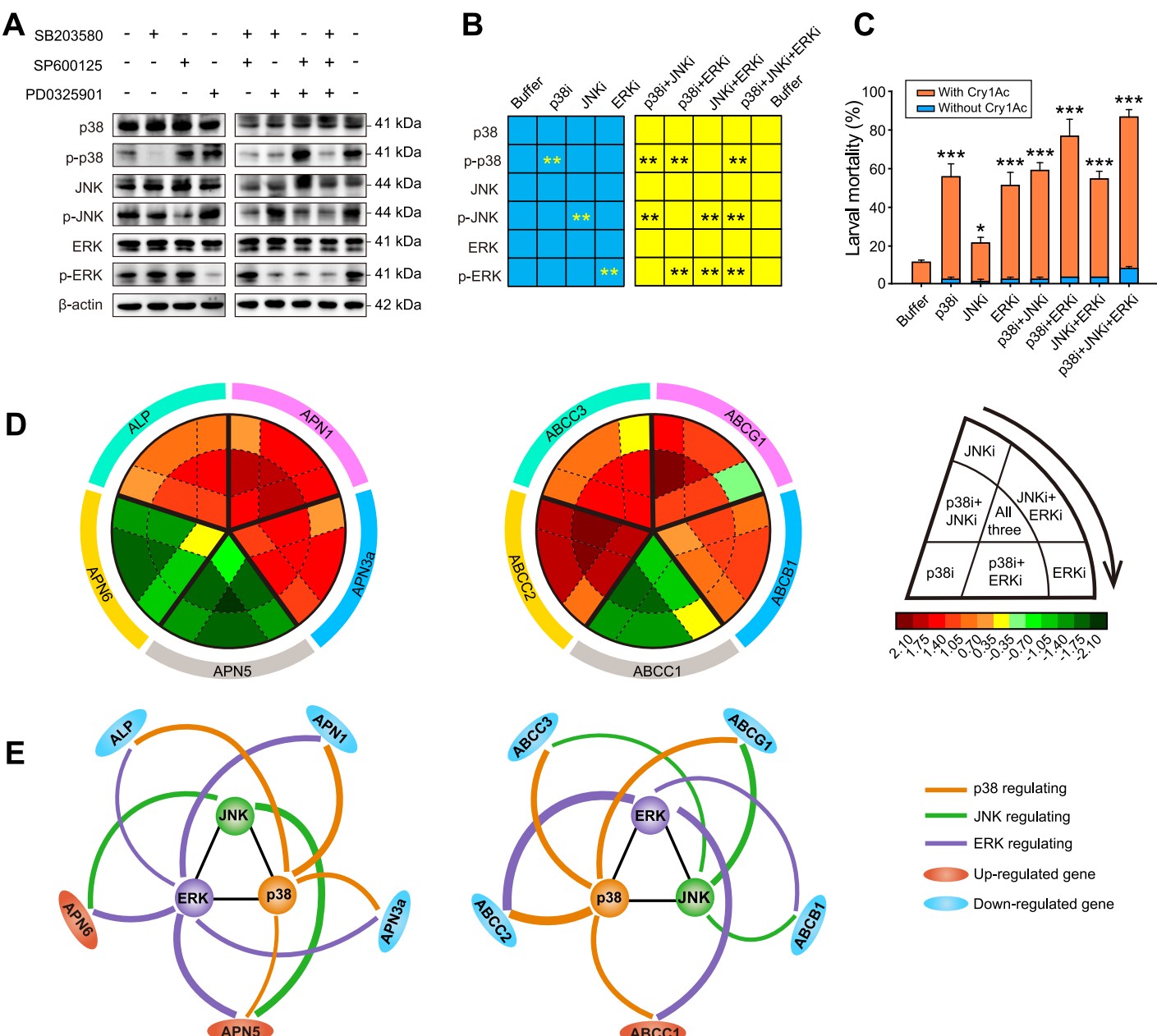

**Fig 4. Functional analysis of p38, JNK and ERK MAPK pathways after treatment with specific inhibitors, in Bt Cry1Ac susceptibility of resistant *P. xylostella*.** (A) Western blot analysis of both total protein and protein phosphorylation levels of p38, JNK and ERK in NIL-R larvae pretreated with specific inhibitors. The β-actin protein was analyzed as an internal loading control. (B) The quantification of representative blots (A) using ImageJ 1.51 from three biological replicates. $^{**}P < 0.01$ by Duncan's test compared to the control. (C) Susceptibility to 1000 mg/L Bt Cry1Ac protoxin in NIL-R larvae treated with the different inhibitors. Data in figures are means and standard errors from three biological replicates. $^{*}P < 0.05$, $^{**}P < 0.01$ and $^{***}P < 0.001$ by Duncan's test compared to the control. (D) qPCR analysis of Cry1Ac resistance-related genes (ALP, APNs and ABC transporters) in midgut tissues of NIL-R larvae pretreated with the indicated inhibitors. The expression levels were transformed to log2 values. All the expression data are from three biological replicates and four technical repeats. (E) Schematic summaries of the regulation effects of p38, JNK and ERK pathways on Cry1Ac resistance-related genes. The thickness of each line is proportional to the strength of the regulation effect. Only regulation effects with significant differences between inhibitor and buffer are shown. The target midgut resistance-related genes with red color indicated that they were up-regulated, while those with blue color indicated that they were down-regulated.

responses systems [3,44]. After the perception of PAMPs, plants can also activate MAPK signaling cascades to confer resistance to both fungal and bacterial pathogens [45]. The work reported here, along with previous published results [30,32], demonstrate that MAPK

signaling cascades are activated at the levels of expression and phosphorylation in different Bt toxin resistant populations of insects and nematodes [39,40,46,47]. In mammalian cells, MAPK signaling pathways were shown to be activated by, and involved in the defense against, numerous bacterial pore-forming toxins [9]. Collectively, these examples indicate that the activation of MAPK signaling cascades as a defense strategy against pathogens or their virulence factors is evolutionarily conserved among diverse organisms.

Various pathogens manipulate MAPK signaling cascades in order to circumvent, suppress or modify immune responses to facilitate their infection [3]. Effectors produced by pathogens, injected into plant or animal cells, have acetyltransferase or phosphothreonine lyase activities that suppress MAPK signaling via blocking phosphorylation of kinases [48,49]. For example, pneumolysin (PLY), a pore-forming toxin produced by *Streptococcus pneumoniae*, hijacks a host factor to inhibit p38 signaling pathway and promote bacterial invasion [9]. While interfering with MAPK signaling cascades is a feature of some pathogens, there are others such as human immunodeficiency virus and silkworm nucleopolyhedrovirus that intentionally trigger a MAPK signaling pathway to enhance their replication and infection [50,51].

Here we demonstrated the role and topology of the MAPK cascades in defending *P. xylostella* against Bt Cry toxins action (Fig 5). We have recently shown that increased titers of 20-hydroxyecdysone (20E) and juvenile hormone (JH) are involved in transducing the upstream signal triggered by the toxin action via MAP4K4 [32] that activated downstream effector responses through three separate MAPK pathways (ERK, JNK and p38), finally resulting in the down-regulation of Cry toxin midgut receptors (ALP, APN1, APN3a, ABCB1, ABCC2, ABCC3 and ABCG1) and the concurrent up-regulation of their non-receptor paralogs (APN5, APN6 and ABCC1), finally suppressing the toxin activity and binding to confer Bt Cry1Ac resistance while retaining cell homeostasis in *P. xylostella* [30–34]. Recently, we found that the MAPK-directed activation of the transcription factor *CREB* leads to P450-mediated imidacloprid resistance in *Bemisia tabaci* [52]. Similarly, it is expected that specific transcription factors (TFs), like Jun and Antp, could serve as downstream effectors and provide a link between the MAPK cascades and expression of specific midgut genes [53–55]. Interestingly, the p38 and ERK signaling pathways selectively regulate certain midgut genes but not others. These regulated midgut genes might contain similar transcription factor binding sites (TFBSs) that can be recognized and regulated by a key MAPK-responsive TF, while other unregulated ones contain distinct functional TFBSs in their promoter regions responding to specific MAPK-responsive TFs. Further studies to identify the functional TFBSs of these midgut genes and the corresponding MAPK-responsive TFs involved in Cry1Ac resistance mechanism in *P. xylostella* are warranted.

Given that the gut tissue is in constant contact with large numbers of pathogens, it must act as a physical barrier armed with efficient systems for pathogen control and cellular homeostasis [7,56]. In the gut defense to pathogens, MAPK signaling pathways have been shown to regulate not only the local production of reactive oxygen species (ROS) to resist pathogen growth [57], but also gut epithelium regeneration to endure the pathogenesis of infection [58]. With the MAPK cascades being involved in a wide variety of pathogen defense/immune processes, it remains to be established exactly how these particular pathways operate and whether particular scaffold proteins are involved in their signal transduction [59–61].

The fact that there is redundancy within the MAPK signaling pathways (i.e. the involvement of two MAP3Ks and three MAP2K/MAPKs) helps to enhance the biological robustness of the response [62–64]. Functional overlap within the MAPK cascades was also previously observed for the p38 and JNK pathways in the nematode defense response against Cry5B toxin action [47]. MAPK cascade components such as TAO and MAP2K7 appear to have little or no role in Cry1Ac resistance response but potentially. Although TAO shows increased

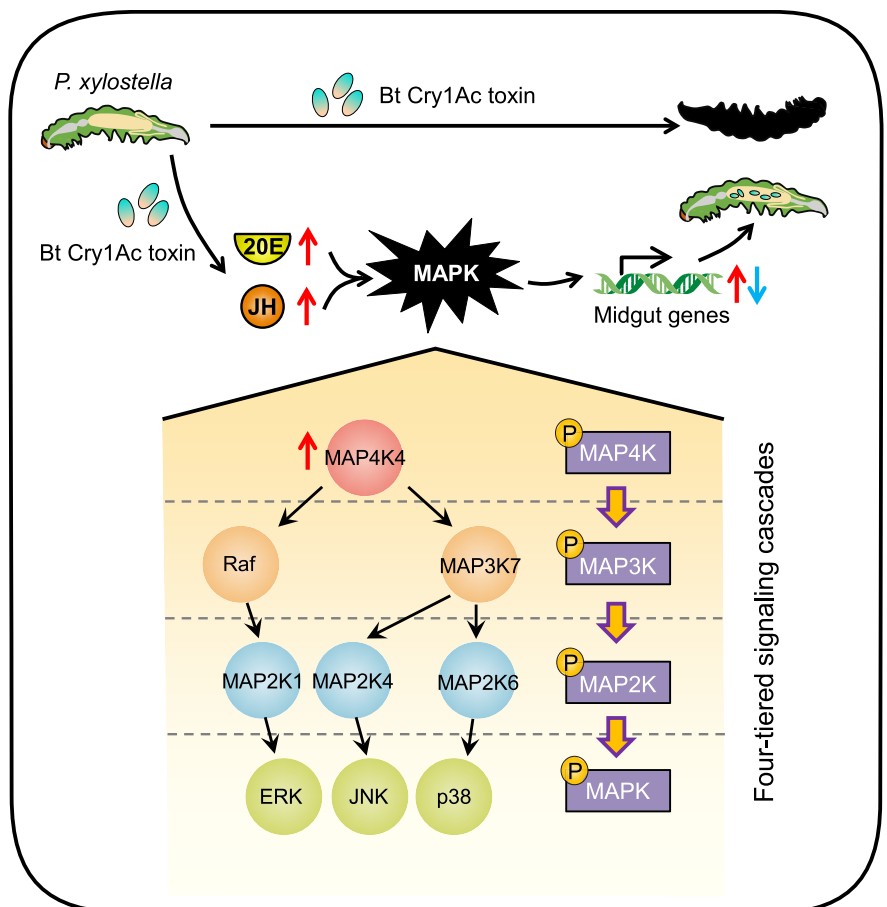

**Fig 5. Schematic representation of the MAPK "road map" for overcoming Bt toxin action in *P. xylostella*.** Upon exposure to Bt Cry1Ac toxin, susceptible *P. xylostella* larvae are killed, whereas with low doses, or in resistant larvae, increased levels of insect hormones (20E and JH) can activate the MAPK signaling cascades to regulate differential expression of multiple midgut genes resulting in larval survival [30,32]. For the road map of MAPK signaling cascades three degenerate, four-tiered, branched signaling pathways, including MAP4K4-Raf-MAP2K1-ERK, MAP4K4-MAP3K7-MAP2K4-JNK and MAP4K4-MAP3K7-MAP2K6-p38, form the modules that regulate this response.

phosphorylation in the resistant strain this could be through a pathway involving MAP4K4 but unrelated to resistance. Our data show that the three MAPK signaling pathways are channeled through MAP4K4. Although this introduces a point of fragility into the network, it also provides a potential target to improve pest control. Given the pivotal role that this kinase plays in the defense against Bt Cry toxins, developing products that impair its function, like RNAi-based pest control strategies or MAP4K4-specific kinase inhibitors, could be used to synergize the action of the toxin and delay the evolution of insect resistance to Cry toxins.

In summary, our data clearly demonstrate that the ERK and p38 pathways play dominant roles in regulating resistance to Bt Cry1Ac toxin in *P. xylostella*, while the JNK pathway plays a secondary role. Global quantitative phosphoproteomic analyses provided clues for identifying crucial MAPK-responsive proteins, such as TFs and scaffold proteins, which are important for signal transduction in Bt Cry1Ac resistance. Moreover, the delineated MAPK roadmap based on the RNAi and inhibitor assays could help inform the development of products to better control this pest.

## Materials and methods

### Insect strains

The five *P. xylostella* strains used in this study have been described in detail elsewhere [65–68]. Briefly, a highly inbred laboratory *P. xylostella* strain DBM1Ac-S was used as the susceptible strain, and was maintained in laboratory conditions without exposure to any Bt toxins or chemical pesticides. Compared to the susceptible DBM1Ac-S strain, the DBM1Ac-R, NIL-R, and SZ-R strains have respectively evolved about 3500-, 4000-, and 450-fold resistance to Cry1Ac protoxin, and the SH-R strain has developed approximately 1900-fold resistance to Bt var. *kurstaki* (Btk) formulation. The field-evolved or laboratory-selected Cry1Ac-resistance in these four independent *P. xylostella* strains have a similar mechanism involving MAPK-mediated differential expression of *PxmALP*, *PxABCB1*, *PxABCC1-3* and *PxABCG1* genes [30,31,33]. These *P. xylostella* strains were fed on Jing Feng No. 1 cabbage (*Brassica oleracea* var. *capitata*) at 25°C with 65% relative humidity (RH) and a 16:8 (light:dark) photoperiod. Adults were supplied with a 10% honey/water solution.

### Toxin preparation and bioassay

As described previously [65,69], the Cry1Ac protoxin was prepared from Btk strain HD-73, and the toxicity of Cry1Ac protoxin to *P. xylostella* larvae was determined in 72 h leaf-dip bioassays. Briefly, ten third-instar *P. xylostella* larvae were used for each of seven toxin concentrations and bioassays were repeated four times. Toxicity bioassay results showed that the resistance ratios of the DBM1Ac-R, NIL-R and SZ-R strains to Cry1Ac protoxin was approximately 3500-, 4000-, and 450-fold that of the DBM1Ac-S strain, respectively, and the resistance ratio of the SH-R strain to the Btk formulation was approximately 1900-fold that of the DBM1Ac-S strain.

### Quantitative phosphoproteomic analysis

To analyze the phosphorylated proteins in *P. xylostella*, the midgut tissue was dissected from fourth-instar larvae of DBM1Ac-S and NIL-R strains and three biological replicates were prepared for each strain. These midgut tissues were homogenized in buffer (40 mM Tris-HCl, 7 M urea, 2 M thiourea, 1% DTT, 1 mM EDTA) supplemented with the EDTA-Free Complete Protease Inhibitor Cocktail (Roche) and the PhosSTOP Phosphatase Inhibitor Cocktail (Roche) according to the manufacturer's instruction, and the homogenates were then sonicated. Lysates were collected by centrifugation at $18000 \times g$ for 40 min at 4°C. The concentrations of midgut proteins were estimated by using the Bradford assay (Biomed). Ammonium bicarbonate was added to 100 μg protein sample to a final concentration of 50 mM. The mixtures were treated with 10 mM DTT at 56°C for 1 h and then alkylated with 55 mM iodoacetamide at room temperature for 40 min in darkness. Trypsin was added to protein samples at a ratio of 1:50 (enzyme-to-substrate, w/w) and incubated overnight at 37°C.

The peptide mixtures were labeled with iTRAQ Reagent-8Plex Multiplex Kit (Applied Biosystems) following the manufacturer's instruction. Three DBM1Ac-S samples were labeled with reagent 118, 119 and 121, while three NIL-R samples were labeled with reagent 114, 116 and 117. Subsequently, phospho-peptides were enriched by titanium dioxide ($TiO_2$) beads. $TiO_2$ beads were preincubated in 200 μl acetonitrile (ACN) and then equilibrated in 200 μl loading buffer (80% ACN, 5% TFA, saturated by glutamic acid). The mixture of labeled phospho-peptides were suspended in 200 μl loading buffer and added to $TiO_2$ beads. The samples were incubated with gentle rotation. After incubation, the beads were washed twice with washing buffer (80% ACN, 5% TFA). The bound phospho-peptides were eluted with 100 μl elution

buffer (15% ammonium hydroxide) three times. Finally, the eluates were collected, dried and prepared for further identification.

Phospho-peptides were then injected into an Eksigent nanoLC 425 system (Applied Biosystems) with a C18 trap column (3 m, 0.10 × 20 mm) and a C18 analytical column (5 m, 0.75 × 150 mm). The mobile phase buffer consisted of buffer A (0.1% formic acid) and buffer B (0.1% formic acid in 80% acetonitrile) under a gradient (5% to 80% of buffer B for 100 min, 80% of buffer B for 10 min, 80% to 5% of buffer B for 0.1 min) at 300 nl/min flow rate. The Eksigent nanoLC 425 system was equipped with a Q-Exactive mass spectrometer (Thermo Fisher Scientific). The mass spectrum was obtained in a data-dependent mode. Full scan MS were performed from *m/z* 350 to 1,750 at a resolution of 70,000 followed by MS/MS scan. AGC target values of MS and MS/MS scans were 3e6 and 2e5, respectively. The dynamic exclusion window was 25 s, while the precursor isolation window was *m/z* 2.0 with normalized collision energy of 28.

The MS/MS raw data were processed using Proteome Discoverer 2.1 (Thermo Fisher Scientific), and subsequently, a database search was conducted using Mascot with an amino acid sequence database generated from the current available transcriptome databases from *P. xylostella* different strains, tissues and ages. The following search criteria were applied: trypsin digestion, up to 2 missed cleavages; carbamidomethyl (C), iTRAQ8plex (N-term) and iTRAQ8plex (K) as fixed modifications; oxidation (M) and phosphorylation (pS/T/Y) as variable modifications; a peptide mass tolerance of 20 ppm, a fragment mass tolerance of 0.1 Da. The false discovery rate of peptide identification was set to 1%. A filter of significantly changed phospho-peptides or proteins were based on Student's t-test (P ≤ 0.05) and fold change (≥ 1.5 or ≤ 0.6667). The final identified phosphorylated polypeptide sequences were listed in S5 Table.

Gene Ontology (GO) enrichment analyses and Kyoto Encyclopedia of Genes and Genomes (KEGG) pathway enrichment analyses were performed in the Gene Ontology resource (http://geneontology.org) and KEGG database (http://www.kegg.jp/), respectively. Protein-protein interaction networks were generated using STRING database (https://string-db.org/) with differentially expressed phosphoproteins and visualized in the Cytoscape software (http://www.cytoscape.org/).

## RNA extraction and cDNA synthesis

The midgut tissues of fourth-instar *P. xylostella* larvae were dissected and homogenized in TRIzol reagent (Invitrogen). Total RNA was extracted following the manufacturer's protocol. After being quantified with a NanoDrop 2000c spectrophotometer (Thermo Fisher Scientific), the first-strand cDNA was synthesized using 5 μg of total RNA with the PrimeScript II 1st strand cDNA Synthesis Kit (TaKaRa) for gene cloning and using 1 μg of total RNA with the PrimeScript RT kit (containing gDNA Eraser, Perfect Real Time) (TaKaRa) for qPCR detection. The synthesized first-strand cDNA samples were stored at -20°C until used.

## MAPK identification and gene cloning

To conduct the genome-wide MAPK gene analysis in *P. xylostella* (S1 Table), the MAPK orthologs of *Homo sapiens*, *Caenorhabditis elegans*, *Drosophila melanogaster* and *Anopheles gambiae* [70] were retrieved from GenBank database (http://www.ncbi.nlm.nih.gov) (S2 Table). These genes were used as queries to screen the Diamondback moth Genome Database (DBM-DB: http://iae.fafu.edu.cn/DBM/). The identified MAPK sequences were further *in silico* corrected by the current GenBank annotation and available *P. xylostella* transcriptome databases. Based on the corrected nucleotide sequences of *P. xylostella* MAPKs, we designed

gene-specific primers with the Primer Premier 5.0 software (Premier Biosoft) to clone their full-length cDNA sequences (S3 Table). PCR reactions (25 μl) were performed in an S1000 Thermal Cycler PCR system (BioRad) using LA Taq polymerase (TaKaRa). The PCR program was as follows: 35 cycles of 94˚C for 30 s, 50–60˚C (depending on the primers) for 45 s and 72˚C for 1–4 min based on the PCR product size; and a final cycle of 72˚C for 10 min. The PCR amplicons with expected size were excised from 1.5% agarose gel and purified using the DNA Purification Kit (CWBIO), and further cloned into pEASY-T5 (Transgen) before introduction into *Escherichia coli* TOP10 competent cells (Transgen) for DNA sequencing. The final cloned full-length cDNA sequences of *P. xylostella* MAPK cascade genes were deposited in the GenBank database (Accession nos. MN211342-MN211357). The deduced MAPK protein sequences were obtained by ExPASy translate tool Translate (http://web.expasy.org/translate/), and submitted to NCBI Batch CD-search database (https://www.ncbi.nlm.nih.gov/Structure/bwrpsb/bwrpsb.cgi) to identify potential MAPK kinase domains. The isoelectric point (pI) and molecular weight (Mw) of these MAPK proteins were calculated in the ExPASy proteomics tool Compute pI/Mw (http://ca.expasy.org/tools/pi_tool.html).

## Phylogenetic analysis

We used amino acid sequences of MAPK orthologs with complete kinase domain from *H. sapiens*, *C. elegans* and 13 arthropod species (S2 Table) to construct a high-quality unrooted phylogenetic tree. All the protein kinase domains of these MAPKs were subjected to sequence alignments through ClustalW using Molecular Evolutionary Genetic Analysis software version 6.0 (MEGA 6.0). Neighbor-joining (NJ) algorithm was used with "p-distance" as amino acid substitution model and "pairwise deletion" as gaps/missing data treatment and 1000 bootstrap replicates.

## Gene selection pressure detection

In the adaptive evolution process of living organisms, protein-coding genes are generally under an active natural selection pressure. Traditionally, this gene selection pressure can be described as the ratio of Ka/Ks: Ka is the rate of non-synonymous substitution and Ks is the rate of synonymous substitution. The Ka/Ks ratio can be adopted to assess either negative or positive selection tendencies for genes of interest. In order to estimate the gene selection pressure, all the MAPKs of 13 arthropod species were analyzed. Based on the protein and cDNA sequence alignments, the Ka, Ks and Ka/Ks values were calculated pairwise with the Ka/Ks_Calculator 2.0 (https://sourceforge.net/projects/kakscalculator2/) using the MYN algorithm. Ka/Ks values < 1, = 1, > 1 indicate genes involved in purifying selection, neutral evolution, and positive selection, respectively.

## qPCR analysis

The detailed procedure used for real-time quantitative PCR (qPCR) analysis to detect gene expression has been described elsewhere [66]. Gene-specific primers of MAPK cascade genes were selected in this study (S3 Table), while qPCR primers of Bt resistance-related genes were obtained from our previous studies [30–33]. The qPCR reactions were performed with 2.5 × SYBR Green MasterMix Kit (TIANGEN) following the manufacturer's instructions in the QuantStudio 3 Real-Time PCR System (Applied Biosystems). Four technical repeats and three biological replicates were conducted for each treatment. Relative expression levels of target genes were calculated using the $2^{-\Delta\Delta Ct}$ method and normalized to the ribosomal protein *L32* (*RPL32*) gene (GenBank accession no. AB180441).

## Western blot

Midgut tissues were dissected from fourth-instar *P. xylostella* larvae of different strains. Midgut tissues were lysed and homogenized in CelLytic M Cell Lysis Reagent (Sigma Aldrich) supplemented with the EDTA-Free Complete Protease Inhibitor Cocktail (Roche) and the PhosSTOP Phosphatase Inhibitor Cocktail (Roche) according to the manufacturer's instruction, and the supernatant containing dissolved midgut proteins was collected by centrifugation. After protein quantification by using the Bradford assay (Biomed), the obtained midgut proteins were separated in 10% SDS-PAGE (CWBIO) with the PageRuler Prestained Protein Ladder (Thermo Fisher Scientific) and electrotransferred to PVDF membranes (Merk Millipore). Membranes were then blocked with Bløk-PO buffer (Merk Millipore) and incubated at 4˚C overnight with specific primary antibodies for the different proteins (S4 Table). HRP-conjugated goat anti-rabbit IgG were diluted 1:5000 and incubated 1 h at 25˚C. Membranes were washed with TBST buffer 4 times for 10 min each time after incubated with antibodies. Blots were revealed by SuperSignal West Pico Chemiluminescent reagent (Thermo Fisher Scientific) and caught by the Tanon-5200 Chemiluminescent Imaging System (Tanon). The images were analyzed using the ImageJ 1.51 software (https://imagej.nih.gov/ij/).

## RNA interference

The dsRNA synthesis and gene silencing were performed as previously described [30,71]. Briefly, gene-specific dsRNA primers harboring a T7 promoter on the 5′ end were used to target different MAPK genes or *EGFP* gene (GenBank accession no. KC896843) were designed using the Primer Premier 5.0 software (Premier Biosoft) (S3 Table). The primer sets used to generate dsRNA of each target *MAPK* gene were designed accordingly to the gene-specific region and not in the conserved kinase domain in order to avoid potential off-target effects, and no specific hits to any other gene were found. The specificity of the selected dsRNA fragments was analyzed by BlastN search on the GenBank and *P. xylostella* genome databases. The obtained PCR products using these gene-specific primers were used as DNA templates for *in vitro* dsRNA synthesis using the T7 RiboMAX Express RNAi System (Promega) following the manufacturer's protocol. The generated dsRNA samples were dissolved in injection buffer [10 mM Tris–HCl (pH 7.0); 1 mM EDTA] and mixed with Metafectene PRO transfection reagent (Biontex) before microinjection into the hemocoels of third-instar NIL-R larvae. A total of 70 nanoliters of buffer, containing dsEGFP (300 ng) or dsRNA (300 ng) were microinjected using a Nanoliter 2000 microinjection system (World Precision Instruments) under an SZX10 microscope (Olympus) with <10% larval mortality 5 days post-injection. More than fifty larvae were microinjected for each treatment and three independent experiments were conducted. The optimal detection time of silencing effect and the quantities of dsRNA injected were optimized in preliminary experiments. Combinatorial RNAi assays involving simultaneous silencing of several MAPK genes were conducted in parallel with single gene RNAi assays. The RNAi effectiveness was validated by qPCR at 48 h post-injection. In addition, to determine the regulation of MAP4K4, MAP3Ks and MAP2Ks on p38, JNK and ERK, both the protein and phosphorylation levels were detected by western blot assays. To access the role of MAPK signaling cascade in regulation of Bt Cry1Ac resistance-related genes in *P. xylostella*, the relative expression level of ALP, APNs and ABC transporters genes were tested by qPCR at 48 h. Leaf-dip bioassays using Cry1Ac protoxin (1,000 mg/L) were performed for 72 h using larvae at 48 h post-injection. Each bioassay replicated three times, larval mortality in control treatments was below 5% and bioassay data processing was as mentioned above.

## MAPK inhibitor assay

To analyze the role of p38, JNK and ERK in regulating Cry1Ac resistance related genes, NIL-R strain larvae were treated with the specific commercial inhibitors of p38, JNK and ERK MAPK. The optimal concentrations and detection time of these specific inhibitors were optimized in preliminary experiments (S8 Fig). After optimization, 30 μM concentration of each SB203580 (specific inhibitor of p38, Merk Millipore), SP600125 (specific inhibitor of JNK, Merk Millipore), and PD0325901 (specific inhibitor of MEK1/2, TargetMol) was selected to conduct these experiments. We used a leaf-dip method similar to the toxicity bioassay. The detailed experimental procedures were as follows. The MAPK inhibitors were first dissolved in DMSO (Sigma Aldrich), then, 0.05% (v/v) Triton X-100 solution was added to the dissolved inhibitors with the final concentration of DMSO as 0.1%. Afterward, the leaf discs (10 cm in diameter) were dipped into inhibitor solutions or DMSO solution alone (control), the leaf discs were air-dried and placed in glass dishes containing wet filter paper. Fifty third-instar NIL-R larvae were tested on the leaf discs. After testing for 6 h, some larvae were used for midgut dissection to obtain RNA and protein samples for qPCR and western blot analysis, respectively. The remaining larvae were used for leaf-dip bioassays as described above.

## Statistical analyses and data visualization

The gene cluster analyses were performed with Cluster 3.0 software (https://www.geo.vu.nl/~huik/cluster.htm) and heat maps were visualized by the TreeView software (https://treeview.co.uk/). For qPCR, western blot and bioassay data, one way ANOVA with Duncan's test were used for analyses of statistical significance ($P < 0.05$) using IBM SPSS Statistics 23.0 (https://www-01.ibm.com/support/docview.wss?uid=swg24038592). Graphs were generated by SigmaPlot 12.5 (https://systatsoftware.com/products/sigmaplot/), GraphPad Prism 7.0 (https://www.graphpad.com/scientific-software/prism/) or R version 3.4.3 (https://www.r-project.org/), and optimized in Adobe Illustrator CC 2015 (www.adobe.com/Illustrator). The raw data of the figures and statistical analyses in this study are provided in S1 Data.

## Supporting information

**S1 Fig. Cloning and Characterization of all identified MAPK cascade genes in *P. xylostella.*** (A) Amplification of full-length cDNA of all identified MAPK cascade genes in *P. xylostella*. M1 and M2 represent two molecular size markers. Lanes 1 to 13 are *PxMAP4K3*, *PxMAP3K4*, *PxMAP3K7*, *PxMAP3K10*, *PxMAP3K12*, *PxMAP3K15*, *PxRaf*, *PxTAO*, *PxMAP2K1*, *PxMAP2K4*, *PxMAP2K6*, *PxMAP2K7* and *PxMAPK15* respectively. Lanes 14 to 16 are *Pxp38*, *PxERK* and *PxJNK*. All the PCR products were resolved by 1.5% agarose gel electrophoresis. (B) Gene structure of all identified MAPK cascade genes. The boxes represent exons and are drawn to scale. The numbers in boxes indicate the length of exons and the numbers above boxes indicate the exons order.
(PDF)

**S2 Fig. Scaffold location of all identified MAPK cascade genes in the *P. xylostella* genome.** The length of scaffolds and the location of MAPK genes are drawn to scale. The sequences on the same scaffold can be assembled to the same gene. For example, Px016598, Px016599 and Px016600 can be assembled to *PxMAP3K12*. The gene sequence of *PxMAP2K1* can be found in the *P. xylostella* genome, but its scaffold location information is absent.
(PDF)

**S3 Fig. Pairwise comparisons of primary sequence identities among MAPK cascade kinases of *P. xylostella.*** Values in each rectangle represent the percent identity of pairs of MAPK

cascade kinases. Gene names are at the left and across the top. Percentage identity for each comparison is color-coded according to the gradient value at the bottom.
(PDF)

**S4 Fig. Distribution of MAPK cascade genes among 15 lepidopteran insects.** The presence or absence of the MAPKs has been assessed in the genome of 15 lepidopteran insects. The gene categories based on the classification of kinase domains are at the top.
(PDF)

**S5 Fig. RNA-seq analysis of MAPK cascade genes in *P. xylostella*.** The RNA-seq data used here were downloaded from the Sequence Read Archive (SRA). (A)–(C) The log2 TPM values of genes were used to create the heatmap by Cluster 3.0. with correlation (uncentered) distance and complete linkage. Heatmaps were visualized by TreeView. (A) Expression patterns (log2 TPM values) of MAPK cascade genes in four developmental stages. E, egg (SRX056231); L, larva (SRX056232); P, pupa (SRX056233); A, adult (SRX056234). (B) Expression patterns (log2 TPM values) of MAPK cascade genes in six adult tissues. AH, adult head (SRX1984133); AA, adult abdomen (SRX1977074); APG, adult pheromone gland (SRX1984138); AL, adult leg (SRX1984145); MA, male antennae (SRX1984140); FA, female antennae (SRX1984104). (C) Expression patterns (log2 TPM values) of MAPK cascade genes in a study of fungal pathogen infection. 24c, 24 h control (SRX1165822); 24t, 24 h infection (SRX1165825); 36c, 36 h control (SRX1165823); 36t, 36 h infection (SRX1165826); 48c, 48 h control (SRX1165824); 48t, 48 h infection (SRX1165827). (D) The absolute expression levels of MAPK cascade genes in midgut tissues of third-instar DBM1Ac-S larvae as determined by the RPKM values of our previous transcriptome and RNA-seq data. The unigenes of MAPK cascade genes were identified by searching against the midgut transcriptome with the full-length cDNA sequence as queries. The RPKM values of these unigenes derived from the RNA-seq libraries were for gene expression analysis.
(PDF)

**S6 Fig. The spatial-temporal expression pattern of Pxp38, PxJNK and PxERK genes in the susceptible DBM1Ac-S *P. xylostella* as determined by qPCR analysis.** Data in the figures are means and stand errors from three biological replicates. Different letters indicate significant differences between different treatments ($P < 0.05$; Duncan's test; n = 3). Developmental stages: EG, egg; L1, first-instar larvae; L2, second-instar larvae; L3, third-instar larvae; L4, fourth-instar larvae; PP, pre-pupae; P, pupae; MA, male adults; FA, female adults. Tissues: HD, head; IN, integument; MG, midgut; TS, testis; MT, Malpighian tubules.
(PDF)

**S7 Fig. Quantification of the relative expression of p38, JNK and ERK MAPK in western blots (Fig 3B) after RNAi of the different MAPK proteins.** Western blot assays were analyzed with ImageJ 1.51. Data in figures show means and standard errors from three biological replicates. Different letters indicate significant differences between different treatments ($P < 0.05$; Duncan's test; n = 3).
(PDF)

**S8 Fig. Optimization of MAPK inhibitors concentration and detection time.** The SB203580, SP600125 and PD0325901 are the specific inhibitors of p38, JNK and ERK, respectively. To determine the appropriate concentrations of each inhibitor, the third-instar NIL-R larvae were treated with different concentrations as follows: 0, 4, 10, 30 μM for 6 h. Based on these results, the proper concentrations of inhibitors selected to be used in the subsequent assays was 30 μM for the three SB203580, SP600125 and PD0325901 inhibitors. To determine

the appropriate detection time, the third-instar larvae were treated with 30 μM of each inhibitor for 0, 3, 6, 12, 24 h. All inhibitors can significantly reduce the phosphorylation level of these kinases after 6 h of treatment, which was selected as the appropriate detection time.
(PDF)

**S9 Fig. Quantification of the relative abundance of p38, JNK and ERK MAPK in western blots (Fig 4A) after inhibition of the different key MAPK proteins.** Western blot assays were analyzed with ImageJ 1.51. Data in figures show means and standard errors from three biological replicates. Different letters indicate significant differences between different treatments (P < 0.05; Duncan's test; n = 3).
(PDF)

**S1 Table. Genome-wide characterization of the MAPK cascade genes in *P. xylostella*.**
(DOCX)

**S2 Table. List of the current available MAPK cascade genes in different species.**
(DOCX)

**S3 Table. Primers used in this study.**
(DOCX)

**S4 Table. Primary antibodies used in this study.**
(DOCX)

**S5 Table. The final identified phosphorylated polypeptide sequences in the quantitative phosphoproteomic analysis.** The polypeptides of MAPK proteins have been marked out in the right side of the table.
(XLSX)

**S1 Data. Raw data used in the figures and statistical analyses.**
(XLSX)

## Author Contributions

**Conceptualization:** Zhaojiang Guo, Shi Kang, Neil Crickmore, Xuguo Zhou, Youjun Zhang.

**Funding acquisition:** Zhaojiang Guo, Shi Kang, Youjun Zhang.

**Investigation:** Zhaojiang Guo, Shi Kang.

**Resources:** Qingjun Wu, Shaoli Wang.

**Supervision:** Zhaojiang Guo, Youjun Zhang.

**Writing – original draft:** Zhaojiang Guo, Shi Kang.

**Writing – review & editing:** Zhaojiang Guo, Shi Kang, Neil Crickmore, Xuguo Zhou, Alejandra Bravo, Mario Soberón, Youjun Zhang.

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
