## [Decision Letter · Decision Letter 0]

12 Jul 2021

Dear Dr. Zhang,

Thank you very much for submitting your manuscript "The regulation landscape of MAPK signaling cascade for thwarting Bacillus thuringiensis infection in an insect host" for consideration at PLOS Pathogens. As with all papers reviewed by the journal, your manuscript was reviewed by members of the editorial board and by several independent reviewers. The reviewers appreciated the attention to an important topic. Based on the reviews, we are likely to accept this manuscript for publication, providing that you modify the manuscript according to the review recommendations. Please address all of the reviewers comments in full.

Sincerely,

Rachel M McLoughlin, PhD

Associate Editor

PLOS Pathogens

Michael Otto

Section Editor

PLOS Pathogens

Kasturi Haldar

Editor-in-Chief

PLOS Pathogens

orcid.org/0000-0001-5065-158X

Michael Malim

Editor-in-Chief

PLOS Pathogens

orcid.org/0000-0002-7699-2064

Reviewer Comments (if any, and for reference):

Reviewer's Responses to Questions

**Part I - Summary**

Reviewer #1: In Gou et al., the authors characterize the MAPK signaling cascades that lead to resistance to the Bacillus thuringiensis (Bt) toxin Cry1Ac in the non-model lepidopteran pest Plutella xylostella, the first one for which Bt resistance was reported in the field. They achieved this by mining available genomic and transcriptomic datasets to identify 17 genes encoding for MAPK cascade members in P. xylostella, many of which they found to be overexpressed in P. xylostella strains resistant to Cry1Ac compared to a susceptible strain. The authors also performed quantitative phosphoproteomic profiling of differentially phosphorylated proteins between the midguts of the Cry1Ac resistant P. xylostella strain MIL-R and the Cry1Ac susceptible P. xylostella strain DBM1Ac-S. They found that the phosphorylated versions of the MAPK cascade members MAP3K7, ERK, TAO, MAP2K6 and p38 were significantly more abundant in the resistant strain, compared to the susceptible strain. To elucidate the topology of the MAPK signaling cascades in P. xylostella, the authors used RNAi of 11 MAPK genes and assessed both the phosphorylation of the downstream MAP kinases p38, JNK and ERK by western blot and the susceptibility of larvae to Cry1Ac exposure. These results allowed the authors to reconstruct the topology of the MAPK cascades involved in P. xylostella resistance to Cry1Ac. The authors also showed that treatment with MAPK inhibitors leads to increase in susceptibility to Cry1Ac.

The authors provide exciting new data about the topology of MAPK signaling cascades that contribute to Cry1Ac resistance in the non-model pest P. xylostella. This is novel for this species, however the topology of MAPK signaling is well described in mammals, Drosophila and C. elegans and these known topologies are like what the authors report for P. xylostella. They also show that MAPK inhibitors can restore susceptibility to Cry1Ac in resistant strains, however they don’t provide further evidence of how this could be used in the field to combat pest resistance to Bt. In general, the study presents novel data and an excellent execution and can be recommended for publication in PLoS Pathogens.

Reviewer #2: The authors made an exhaustive study on the MAPK signaling pathways which, in turn, it was previously known that their regulation was altered in some Bt resistant strains. The study shows which pathways are more important determining the susceptibility of P. xylostella to Bt proteins.

Reviewer #3: In this article, the authors conducted a genome-wide identification and characterization of the MAPK signaling cascades and established an MAPK “road map” in the diamondback moth Plutella xylostella. They identified a total of 17 MAPK orthologs in P. xylostella and cloned these MAPK genes. They showed that the transcription levels of most select MAPK cascade genes were up-regulated in the midgut of all resistant P. xylostella strains. They also showed that the phosphorylated protein levels of p38, JNK and ERK were increased in the resistant strains, and that MAP2K6 was involved in activation of p38, MAP2K1 in activation of ERK, and both MAP2K4 and MAP2K7 in activation of JNK. Their data also confirmed that MAP4K4 is the key kinase in activation of all three MAPKs. These results were confirmed by bioassays through RNAi experiment and specific kinase inhibitor assays. Overall, the authors showed that MAPK signaling pathways are involved in regulation of Cry1Ac toxin receptors (ALPs, APNs and ABC transporters), resulting in toxin resistance in P. xylostella. Since p38 pathway can regulate select toxin receptors (for example, APN1, APN3a, APN5, ABCC1, ABCC2 and ABCC3) but not some other receptors (APN6 and ABCB1), and similarly, ERK pathway can regulate select toxin receptors but not some other receptors, it would be great if the authors can elaborate on the similarity and difference of the transcription factor binding sites in the promoter regions of these receptor genes.

PLOS authors have the option to publish the peer review history of their article (what does this mean?). If published, this will include your full peer review and any attached files.

Reviewer #1: No

Reviewer #2: No

Reviewer #3: No

**Part III – Minor Issues: Editorial and Data Presentation Modifications**

Reviewer #1: Major Comments

1. On page 5 of 36 lines 97-99 the authors write: “The fact that resistance in P. xylostella has been linked with multiple and different mechanisms has motivated us to further gain a clear understanding of the underlying interactions, especially for the new-found MAPK signaling pathway” -I would not claim that the MAPK is new-found, this is misleading as much is known about MAPK signaling in response to pore-forming toxins in mammals, Drosophila and C. elegans (which should be mentioned in the introduction to give the reader a broad overview of what is known prior to this paper.) The authors should also emphasize more that MAPK signaling pathways had already been identified to be important in the response against Cry toxins in insects (referencing e.g. previous publications by the authors, https://doi.org/10.1038/srep43964, etc…), but here they explore the full repertoire and function for this particular species (P. xylostella).

2. Figure 5 contains some small unlabeled spheres in different tiers of the MAPK signaling pathways, that suggests that the authors suspect that there are additional MAPK cascade genes that they did not identify. The authors should discuss whether this is the case, why their methodology did not pick up these additional members and what could be done in the future to identify them.

3. Using quantitative phosphoproteomic profiling, the authors found that the TAO MAP3K was significantly more phosphorylated in the resistant strain compared to the susceptible strain, however their genetic analyses revealed that TAO had no role in the resistance of P. xylostella to Cry1Ac. Could the authors discuss in more detail why this could have been and also why they didn’t find more MAPK cascade members significantly more phosphorylated in the resistant strain compared to the susceptible strain?

Minor Comments

1. In Fig 1A the authors could include the data from Homo sapiens and C. elegans that they include in Table S2 for reference.

2. On page 6 of 36 line 120, the authors claim that the Ka/Ks values calculated for MAPK members reflects low selection pressure. I would disagree with this as values between 0 and 0.25 imply a clear trend towards purifying or stabilizing selection, or selection pressure acting against change in these sequences.

3. In figure 1C the authors should indicate which branches correspond to MAP3K.

4. In Fig S5C there is not so much difference in expression of MAPK cascade genes in fungal infection compared to control. The authors could represent this as Log2Fold Change expression relative to control to highlight this.

5. Is there no data of P. xylostella exposure to Cry1Ac to include in Fig S5C? This would be more relevant for the purpose of the paper.

6. On page 7 of 36 line 146, the authors should write “(…) genes were up-regulated in the midgut tissues of all resistant strains compared to the susceptible strain DBM1Ac-S.” This enhances clarity for the reader.

7. The legend of figure 1 should clarify that panels D-F quantify basal expression of MAPK cascade genes without Cry1Ac exposure.

8. The legend of figure 2 should clarify that it presents basal differentially phosphorylated proteins between resistant and susceptible strains without Cry1Ac exposure. Also, it should mention that the protein-protein interaction data comes from the STRING database.

9. In S8 Fig, at what time were the samples collected to determine the MAPK inhibitor concentration?

10. The legend of Figure 5 should include the citation of the paper that reports that insect hormones 20E and JH can activate MAPK signaling, since this was not described in the current work.

**Part II – Major Issues: Key Experiments Required for Acceptance**

Reviewer #2: The final conclusion of the study is that the MAPK signaling pathways play an important role in the susceptibility of P. xylostella to Bt proteins. The authors show upregulation and downregulation of some of the genes in the resistant strains. However, it is not clear how this regulation would interfere with the toxic action of the Cry proteins from Bt. In other words, would the increase in the final gene products (membrane receptors) confer resistance to Bt? Or is the depletion of such gene products what would confer resistance? This aspect is not clear in the final conclusions. I think the authors should discuss on this.

Figure Files:

Data Requirements:

Reproducibility:

References:

---

## [Decision Letter · Decision Letter 1]

25 Aug 2021

Dear Dr. Zhang,

We are pleased to inform you that your manuscript 'The regulation landscape of MAPK signaling cascade for thwarting Bacillus thuringiensis infection in an insect host' has been provisionally accepted for publication in PLOS Pathogens.

Best regards,

Rachel M McLoughlin, PhD

Associate Editor

PLOS Pathogens

Michael Otto

Section Editor

PLOS Pathogens

Kasturi Haldar

Editor-in-Chief

PLOS Pathogens

orcid.org/0000-0001-5065-158X

Michael Malim

Editor-in-Chief

PLOS Pathogens

orcid.org/0000-0002-7699-2064

Reviewer Comments (if any, and for reference):

Reviewer's Responses to Questions

**Part I - Summary**

Reviewer #1: I am satisfied with the authors' responses and recommend the paper for publication.

Reviewer #2: As stated in my previous review of this manuscript, the results are of great interest and novelty

Reviewer #3: The authors have addressed this reviewer's concern.

PLOS authors have the option to publish the peer review history of their article (what does this mean?). If published, this will include your full peer review and any attached files.

Reviewer #1: No

Reviewer #2: No

Reviewer #3: No

**Part II – Major Issues: Key Experiments Required for Acceptance**

Reviewer #1: (No Response)

---

## [Editor Report · Acceptance letter]

1 Sep 2021

Dear Dr. Zhang,

We are delighted to inform you that your manuscript, "The regulation landscape of MAPK signaling cascade for thwarting Bacillus thuringiensis infection in an insect host," has been formally accepted for publication in PLOS Pathogens.

Best regards,

Kasturi Haldar

Editor-in-Chief

PLOS Pathogens

orcid.org/0000-0001-5065-158X

Michael Malim

Editor-in-Chief

PLOS Pathogens

orcid.org/0000-0002-7699-2064